# Microbial Production of Human Milk Oligosaccharides

**DOI:** 10.3390/molecules28031491

**Published:** 2023-02-03

**Authors:** Dileep Sai Kumar Palur, Shannon R. Pressley, Shota Atsumi

**Affiliations:** Department of Chemistry, University of California, Davis, Davis, CA 95616, USA

**Keywords:** human milk oligosaccharides, synthetic biology, metabolic engineering

## Abstract

Human milk oligosaccharides (HMOs) are complex nonnutritive sugars present in human milk. These sugars possess prebiotic, immunomodulatory, and antagonistic properties towards pathogens and therefore are important for the health and well-being of newborn babies. Lower prevalence of breastfeeding around the globe, rising popularity of nutraceuticals, and low availability of HMOs have inspired efforts to develop economically feasible and efficient industrial-scale production platforms for HMOs. Recent progress in synthetic biology and metabolic engineering tools has enabled microbial systems to be a production system of HMOs. In this regard, the model organism *Escherichia coli* has emerged as the preferred production platform. Herein, we summarize the remarkable progress in the microbial production of HMOs and discuss the challenges and future opportunities in unraveling the scope of production of complex HMOs. We focus on the microbial production of five HMOs that have been approved for their commercialization.

## 1. Introduction

Carbohydrates are the most abundant biomolecules on earth and also are the primary energy source for diverse organisms [1]. Human milk, the primary source of nutrition for infants, contains about 7% carbohydrates [2]. Lactose is the primary sugar component in human milk. In addition to lactose, various non-lactose carbohydrates, called human milk oligosaccharides (HMOs) have been identified [3]. Currently, over 200 different HMO structures are known, and with development of cutting-edge analytical methods, more HMO structures are being elucidated [4]. HMOs are made of five basic monosaccharides: D-glucose, D-galactose, N-acetylglucosamine (GlcNAc), L-fucose, and N-acetylneuraminic acid (Figure 1) [5]. Almost all HMO contain a lactose motif at the reducing end, which can be elongated by either (β1,3) or (β1,4) glycosidic bonds [6]. The addition of L-fucose and N-acetyl neuraminic acid to these linear or branched chain oligosaccharides further gives rise to the diverse oligosaccharide structures observed in human milk [5,6]. These complex oligosaccharides contribute significantly to the health of infants by lowering their susceptibility towards diseases [3].

Therefore, human milk is recognized as the gold standard for human infant nutrition by the World Health Organization and the United Nations Children’s Fund [6]. It consists of all the nutritional and physiological components required for the first 6 months of an infant’s life [7]. HMOs possess prebiotic, immunomodulatory, and antagonistic properties towards pathogens and can also serve as a key support for the neurocognitive development of infants [8,9,10]. These and other potential benefits of HMOs make them attractive research targets for preventing or treating diseases in both children and adults [8]. The bioactive properties of HMOs have prompted efforts to elucidate the mechanisms of the individual HMO’s actions [8,9,10], but the sources of HMOs for such studies are limited. Due to limited access to these HMOs, there is an emerging need to synthesize these complex molecules [10].

## 2. HMO Production Methods

The global HMO market was valued at USD 199 million in 2022 [11]. Lower prevalence of breast-feeding around the world [12] and rising popularity of the concept of nutraceuticals has contributed to the growing market size of HMO as a food additive [13], leading to increased demand for a cheap, large-scale production platform for synthesizing HMOs.

Several successful strategies have been developed to chemically synthesize HMOs. One of the earliest chemical syntheses was developed for Lacto-N-tetraose (LNT) in 1979 [14]. Advancement in solid phase synthesis, involving the attachment of sugar molecules to solid support, enabled the synthesis of the complex hexose sugar lacto-N-hexaose (LNH) [15]. Recently, gram-scale production of LNT was achieved from cheap starting materials such as lactose, D-glucosamine, and D-galactose, which paved the way for understanding and elucidating the importance of this sugar in human milk [16]. Over the years, more than 15 different HMOs have been chemically synthesized [3]; however, the major drawback in the chemical synthesis of HMOs is the need for many protecting group manipulations, which increase with the chain length of the target oligosaccharide [17]. Poor atom economy increases the cost of reagents, and a large number of intermediate purification steps leads to low product yields [18]. Therefore, most chemical syntheses of HMOs are not cost-efficient.

Chemoenzymatic synthesis has emerged as an alternative to traditional chemical synthesis methods for the production of natural products [19]. This approach combines the flexibility of chemical synthesis methods with the high regio- and stereoselectivity of enzyme-catalyzed reactions, eliminating the need for protection and de-protection steps [20]. Recently, 31 HMOs were chemoenzymatically synthesized [21].

Building upon these findings, a multigram-scale production system method was established for LNT synthesis based on the one-pot multienzyme (OPME) strategy [22]. OPME involves performing multistep enzymatic reactions in a single reaction system thereby eliminating the need to isolate reaction intermediates [23]. The number of diverse HMOs made available through chemical and chemoenzymatic synthesis is unparalleled [24]. Nevertheless, the high cost of purified enzymes, as well as the necessary addition of similarly expensive cofactors such as ATP, makes chemoenzymatic synthesis a less economically feasible strategy for the industrial-scale production of HMOs [25].

Microorganisms can be used as whole-cell catalysts for the production of industrially relevant molecules [26,27]. Microbial cell factories utilize cheap carbon feedstocks and also possess the high specificity of biocatalysts, which makes them great candidates for industrial-scale production [28]. In addition, microbial cell factories function at relatively lower temperatures and do not require costly purified enzymes and cofactors [28]. Recent progress in synthetic biology and metabolic engineering tools have enabled microbial production platforms to be an alternative and sustainable source of chemicals [29]. About 42 HMO structures have been produced using microbial cell factories [13]. Escherichia coli has emerged as the preferred microbial host for HMO synthesis owing to its fast growth, genetic tractability, and ease of scaling up culture volumes [30]. E. coli does not possess any native glycosyltransferase enzymes to produce HMOs. However, the presence of an efficient pathway for sugar nucleotide production, a precursor for HMO biosynthesis [31,32], makes this organism a viable platform for HMO synthesis. In this review, we summarize the latest progress in the microbial production of HMOs and discuss the challenges and future opportunities in unraveling the scope of production of complex HMOs. Herein, we focus on the microbial production of five HMOs that are approved for their commercialization and several companies are currently developing their industrial-scale production methods [30].

## 3. HMO Production by Microbial Cell Factories

### 3.1. 2′-Fucosyllactose

2′-Fucosyllactose (2′-FL) is the most abundant HMO present in human milk (Figure 1) [33]. The abundance of 2′-FL has attracted a lot of attention to its potential activity [34]. 2′-FL acts as prebiotic, promotes early development of healthy Bifidobacteria-dominated gut microbiota, and provides protection against specific diarrheal diseases in infants [35]. The regulatory approval regarding the commercialization of 2′-FL produced by microbial cell factories was granted in 2016 [13]. Recently, 2′-FL has been produced on a multigram scale via chemoenzymatic and chemical synthesis strategies [36,37]. However, there are various challenges to commercializing these methods, leading to the development of inexpensive and efficient systems to produce 2′-FL.

The biosynthetic pathway of 2′-FL involves the addition of GDP-L-fucose onto the galactose moiety of lactose by an (α1,2)-fucosyltransferase. Two metabolic pathways have been reported to generate GDP-L-fucose, namely the de novo pathway and the salvage pathway [38,39]. The de novo pathway is ubiquitous in both prokaryotes and eukaryotes [38]. The conversion from fructose-6-phosphate to GDP-L-fucose is carried out in five enzymatic reactions (Figure 2) [38]. Mannose-6-phosphate isomerase, ManA, converts fructose-6-phosphate to mannose-6-phosphate followed by the conversion of mannose-6-phosphate to mannose-1-phosphate by phosphomannomutase, ManB; α-D-mannose-1-phosphate guanyltransferase, ManC, converts mannose-1-phosphate to GDP-D-mannose (Figure 2). GDP-D-mannose is then converted to GDP-4-keto-6-deoxymannose by GDP-D-mannose-4,6-dehyratase, Gmd, and finally, GDP-L-fucose is produced from GDP-4-keto-6-deoxymannose by GDP-L-fucose synthetase, WcaG [38]. The salvage pathway, generally found in eukaryotes, has also been recently discovered in the gut bacterium *Bacteroides fragilis* [39]. This pathway involves a bifunctional fucokinase/fucose-1-phosphate guanylyltransferase (Fkp), which converts exogenous fucose to GDP-L-fucose in a two-step enzymatic reaction (Figure 3) [39]. Fkp converts L-fucose to L-fucose-1-phosphate followed by conversion of L-fucose-1-phosphate to GDP-L-fucose. For the last step in the 2′-FL biosynthetic pathway, several (α1,2)-fucosyltransferases have been identified [40].

A 2′-FL production system from sucrose as the sole carbon source was developed in *E. coli* (Figure 3) [41]. The key intermediates, lactose, and GDP-L-fucose, were produced from sucrose (Figure 3). (β1,4)-galactosyltransferase, GalTpm1141, from *Pasteurella multocida* showed high specific activity towards the production of lactose from D-glucose and UDP-D-galactose [41]. Chromosomal integration of genes involved in the de novo pathway of GDP-L-fucose and cscABKR from *E. coli* W enabled the engineered strain to grow on sucrose and increase production of substrates, lactose, and GDP-L-fucose, in the biosynthetic pathway. The deletions of *lacZ*, *pfkA*, *gnd,* and *wcaJ* increased the production of 2′-FL. The expression of the TP Y.b. gene encoding a sugar efflux transporter from *Yersinia bercovieri* ATCC43970 allowed the strain to export 2′-FL out of the cell. The engineered strain expressing *wbgL* encoding for α1–2-fucosyltransferase from *E. coli* O126 [42] produced 3 g/L of 2′-FL using sucrose as the sole carbon source in a fed-batch bioreactor. Due to the promiscuity of WbgL, side products 3FL and LDFT were formed. To reduce 3FL production, *afcB* encoding an α1–3-fucosidase from *Bifidobacterium bifidum*, and the salvage pathway gene, *fkp*, were introduced into the production strain. AfcB hydrolyzes 3FL to produce lactose and L-fucose (Figure 3). Fkp converts the free L-fucose to GDP-L-fucose via L-fucose-1-phosphate by the salvage pathway. The strain produced ~60 g/L of 2′-FL using sucrose as the sole carbon source in a fed-batch 3 L fermenter [41].

The biosynthetic pathway of 2′-FL production via the de novo pathway for GDP-L-fucose was developed in *E. coli* (Figure 2) [43]. The overexpression of genes involved in de novo pathway genes using a high-copy-number plasmid was used to enhance GDP-L-fucose production. To improve lactose import, *lacY* was additionally expressed. To reduce lactose degradation, *lacZ* was deleted. The deletion of *wcaJ* blocks metabolic flux from GDP-L-fucose to colonic acid (Figure 2), thereby increasing GDP-L-fucose accumulation. Chromosomal integration of *wbgL* encoding for (α1,2)-fucosyltransferase enabled the strain to produce 2′-FL. The final production strain expressing *wbgL* produced 79 g/L of 2′-FL using glycerol and lactose as carbon sources in a 3 L fermenter [43].

The *futC* gene encoding for (α1,2)-fucosyltransferase from *Helicobacter pylori* [40] was used for 2′-FL production in *E. coli* due to its high specificity towards lactose [44]. The expression levels of genes involved in lactose uptake and de novo L-fucose assimilation were modulated to improve the accumulation of lactose, GDP-L-fucose, and the intermediates for 2′-FL production. The biosynthetic pathway for 2′-FL utilized one NADPH and GTP molecule. To reduce cell burden, the balance of NADPH and GTP was achieved by the overexpression of *zwf*, *pntAB,* and *gsk*, leading to a significant increase in 2′-FL production. The final strain produced 22 g/L of 2′-FL from lactose in a 3 L fermenter [44].

*Saccharomyces cerevisiae* was engineered to produce 2′-FL from xylose and lactose [45]. The use of yeast as a host organism avoids the risk of bacteriophage infection during large-scale production, possible endotoxin contamination, and unfavorable consumer perception. A lactose transporter gene *LAC12* from *Kluyveromyces lactis* and *wbgL* from *E. coli* O126 were introduced into *S. cerevisiae*. To enable the de novo production of GDP-L-fucose, the *gmd* and *wcaG* genes from *E. coli* were introduced to *S. cerevisiae*. The engineered strain produced 26 g/L of 2′-FL from xylose and lactose in a fed-batch 1L fermentor [45]. The putative (α1,2)-fucosyltransferase FutBc from *Bacillus cereus* was also used to produce 2′-FL. Expression of *futBC* enhanced 2′-FL production by 1.8-fold compared to *futC* from *H. pylori*. The *futBC* genes along with *LAC12* was introduced into *S. cerevisiae*. The strain produced 27 g/L of 2′-FL from lactose and sucrose in a fed-batch 5 L bioreactor [46].

### 3.2. LDFT (Lactodifucotetraose)

LDFT is the most abundant fucosylated HMO and over the course of the first year of lactation, LDFT concentration in human milk is ~0.43 g/L [47]. LDFT can be metabolized by immunomodulatory bacteria, promoting healthy gut microbiome in infants [48]. The lack of availability of LDFT is a major roadblock to further understanding its health benefits. The unique challenge posed by the chemistry of LDFT structure, two different glycosidic bonds between fucose and lactose, makes it a difficult molecule to synthesize chemically. LDFT produced by microorganisms was given regulatory approval as Generally Recognized As Safe (GRAS) in 2018 [30]. The presence of a promiscuous enzyme, (α1,3/4)-fucosyltransferase (3/4FT) from *H. pylori* UA948 was successfully isolated and expressed in *E. coli* [49].

*E. coli* was engineered to produce LDFT from lactose and L-fucose [50]. The biosynthetic pathway of LDFT production involves the addition of GDP-L-fucose onto both glucose and galactose moiety of lactose by an (α1,3)-fucosyltransferase and (α1,2)-fucosyltransferase, respectively (Figure 4) [50]. It has been shown that α1–3/4-fucosyltransferase (3/4FT) from *H. pylori* UA948 can use both non-fucosylated and α1–2-fucosylated galactosyl oligosaccharides as substrates [22,51], while for (α1,2)-fucosyltransferases, WbgL is selective towards lactose and other non-fucosylated galactosyl oligosaccharide acceptor substrates [22,42]. For the production of GDP-L-fucose, *fkp* encoding for bifunctional fucokinase/fucose-1-phosphate from *Bacteroides fragilis* was expressed (Figure 4). The *lacZ* and *fucU* genes were deleted to remove competing pathways. To enhance the uptake rate of L-fucose and lactose into the cell, the sugar transporter genes *lacY* and *fucP* were additionally expressed. The final strain produced 5.1 g/L of LDFT from 3 g/L lactose and 3 g/L L-fucose, achieving 91% of the theoretical maximum yield in test tube conditions. The production of monofucoside side products (2′-FL and 3-FL) was very low compared to LDFT, suggesting the potential viability of this production system for large-scale LDFT production [50].

### 3.3. 3′ and 6′-Sialyllactose

Sialylated HMOs are of interest due to their roles as dietary sources of sialic acid, which plays a role in preventing intestinal infections and promoting healthy brain function in humans [52]. As with other HMOs, sialylated HMOs have been investigated as biosynthetic targets due to the prohibitive cost of chemical synthesis, which suffers from low yields due to the many protection and de-protection steps needed to achieve a stereo- and regiospecific product. The sialic acid *N*-acetylneuraminic acid (Neu5Ac) is rare in nature and difficult to chemically synthesize for similar reasons, meaning that an efficient biosynthesis of any sialyllactose must also include the biosynthesis of the Neu5Ac moiety itself (Figure 5). An exogenous pathway was used to biosynthesize CMP-Neu5Ac in combination with a glycosyltransferase to yield 3′-sialolactose [53]. In this study, three exogenous genes (*neuABC*) from *Campylobacter jejuni* ATCC 4348 were expressed in *E. coli* K12, thus completing a pathway from glycolysis to CMP-Neu5Ac (Figure 6). Genes for competing pathways (*nanA*, *nanK*) were deleted to direct carbon flux towards CMP-Neu5Ac. Finally, an (α2,3)-sialyltransferase gene from *Neisseria meningitidis* was introduced to convert CMP-Neu5Ac to 3′-sialyllactose in the presence of lactose, which serves as the donor substrate for the glycosyltransferase enzyme (Figure 5). The engineered strain was grown to high density (OD_600_~100) on mineral media with a continuous supply of glycerol and lactose, producing 15.5 g/L of 3′-sialyllactose (3′-SL) after 70 h in a 2 L fermentor.

Efforts to biosynthesize 6′-sialyllactose (6′-SL) have required more optimization than efforts to biosynthesize 3′-SL due to the relative promiscuity of (α2,6)-sialyltransferase enzymes (Figure 5). The sialyltransferase gene in the 3′-SL-producing strain described above [53] was substituted with an (α2,6)-sialyltransferase from *Photobacterium* sp. JT-ISH-224 [54]. The (α2,6)-sialyltransferase was chosen because it was the most specific (α2,6)-sialyltransferase known at the time [54]. When this strain was grown on glycerol and lactose, it was found that lactose was exclusively conjugated with 2-keto-3-deoxyoctonate (KDO). To combat the apparent promiscuity of the sialyltransferase enzyme, a variety of strategies were used. First, the supply of substrate for the sialyltransferase was increased by placing the *neuABC* operon under a stronger promoter on a plasmid with a higher copy number. With this change in place, KDO-lactose was no longer formed, and the strain yielded the intended product 6′-sialyllactose and the side product 6,6′-disalyllactose (Figure 5). Hypothesizing that this disialylated product was formed by a side activity of the sialyltransferase, the strain was fed a constant supply of lactose to outcompete 6′-SL as a substrate. The strain produced 16 g L^−1^ of 6′-siallylactose after 47 h with a negligible amount of 6–6′-disialyllactose in a 2 L fermentor.

### 3.4. LNT (Lacto-N-Tetraose)

LNT represents a core structure for most HMO structures as most HMOs are derived from different attachments of L-fucose and the sialic acid group (Figure 1) [3]. It is also the most abundant solid component in HMOs, comprising 6% (*w*/*w*) of total HMOs [55]. Efficient one-pot multienzyme (OPME) strategies have been developed for the production of LNT [22,56]. The biosynthetic pathway of LNT involves the production of UDP-N-GlcNAc and the production of UDP-galactose (Figure 6). Both these intermediates are added to a lactose unit to form LNT (Figure 6) [55,57,58,59]. LNT produced by microorganisms was given regulatory approval as GRAS for commercialization in 2016 [13].

Structural elucidation of LNT-like units in the lipopolysaccharides of *Neisseria meningtidis* led to the discovery of (β1,3)-N-acetylglucosaminyltransferase enzyme, LgtA, which can add a galactose unit to lactose to produce LNT II, the precursor of LNT (Figure 6) [60,61]. The O antigen structure of *E. coli* O55:H7 consists of a repeating polysaccharide containing Gal b1,3GlcNAc. The (β1,3)-galactosyltransferase, WbgO, which is involved in forming the repeating polysaccharide, was identified and shown to convert LNT II to LNT [22,62]. *E. coli* K12 was engineered to produce LNT (Figure 6) [57]. To avoid the use of antibiotics and increase gene stability, chromosomal integration of the heterologous glycosyltransferase gene instead of a plasmid expression system was used. The *lgtA* gene encoding for (β1,3) N-acetylglucosaminyltransferase, LgtA, from *Neisseria meningtidis* was integrated into the *lacZYA* locus, thereby eliminating the lactose-utilization pathway. The *wbgO* gene encoding b1–3-galactosyltransferase, WbgO, from *E. coli* O55:H7 was integrated into the *xylAB* locus in *E. coli*. To increase lactose uptake, the *lacY* gene encoding for lactose transporter LacY was additionally expressed under a strong inducible Ptac promoter [63]. In a shaken flask, this strain produced 0.2 g/L of LNT from glucose and lactose [57]. However, the strain produced 9 times more LNT II than LNT. This suggests that the availability of UDP-galactose and the activity of WbgO would be the bottleneck for the LNT production [57]. With galactose instead of glucose, the strain produced 0.8 g/L LNT in a fed batch bioreactor [58]. The presence of glucose inhibits lactose utilization by catabolite repression and inducer exclusion, so galactose is a better feedstock for the production of LNT [64].

To increase the UDP-galactose pool, the genes (*galE*-*galT*-*galK*) of the Leloir pathway [65] were overexpressed (Figure 6). The deletion of *ugd* encoding for UDP-glucose-6-dehyrdogenase has been previously shown to increase UDP-galactose accumulation in *E. coli* [65]. Thus, *ugd* was deleted in the production strain, and the final strain produced 32 g/L of LNT from glycerol and lactose in a fed-batch bioreactor [55].

Several sugar efflux transporters have been identified in *E. coli* [66]. These transporters relieve the osmotic stress due to the accumulation of sugars by exporting sugars out of the cells. The presence of LNT II in the production media suggests that *E. coli* has a transporter for LNT II [59]. Fourteen native sugar transporters were screened to identify an LNT transporter. This study showed that SetA is the major LNT II transporter in *E. coli*. The deletion of *setA* increased LNT titers from 2.19 g/L to 2.96 g/L and exogenous LNT II concentration decreased from 0.52 g/L to 0.23 g/L [59].

## 4. Challenges

When considering the application of whole-cell biocatalysts in the production of food additives, food safety must be thoroughly considered. Food products based on Gram-negative bacteria such as *Escherichia coli* increase the likelihood of lipopolysaccharide contamination [67]. Lipopolysaccharides are a structural component of bacterial cells and are also known as endotoxins [68]. Bacterial lipopolysaccharides have been associated with a number of diseases, including liver damage, neurological degeneration, gut inflammation, and diabetes [67]. The purification of HMOs produced by *E. coli* to remove these endotoxins is a crucial safeguard and will remain a major hurdle for the commercialization of these microbial production platforms [68].

Sugar transporters can be useful tools in enhancing HMO production, but they can also hamper biosynthesis. The presence of several native sugar transporters in *E. coli*, which can export HMO precursors out of the cell, is not optimal for the production of more complex HMOs that have many intermediates [66]. The identification and characterization of these sugar transporters are very important for HMO production not only to minimize substrate loss but to selectively export target HMOs out of the cell. In addition, more complex HMOs may not have any known transporters, and the biosynthesis of these targets could be limited by the buildup of the product in the cytosol. The study of new and existing sugar transporters is necessary to progress the field of HMO biosynthesis.

There is a lack of protein-engineering strategies to increase the activity, specificity, and expression of glycosyltransferases in microbial hosts. Underperformance in these areas leads to low titers and the formation of side products [53,57]. For example, in HMO biosynthetic pathways involving the expression of multiple glycosyltransferases, the use of highly specific enzymes can help in controlling the order of glycosylation and avoiding the formation of side products.

## 5. Conclusions and Perspectives

Progress in the field of analytical methods continues to reveal new molecules contributing to the vast diversity of HMOs. With the increasing beneficial effects of HMOs being reported, there is a growing demand for the large-scale production of HMOs. Here, we summarize the latest research in the microbial production of HMOs. Microbial cell factories function at relatively low temperatures, do not require costly purified enzymes and cofactors, utilize cheap carbon feedstock, and also possess the high specificity of biocatalysts, which makes them great candidates for industrial-scale production. The lack of knowledge on the sugar transporters for HMOs is a major bottleneck in the production of long-chain HMOs. Furthermore, the unintentional export of intermediates in the biosynthetic pathway will lead to low titer and yield. The fundamental basis for metabolically engineering HMO-producing strains is the activity of glycosyltransferase enzymes, and therefore, there is a need for protein engineering to increase the specificity and solubility of these heterologous genes in microbial production hosts. Nevertheless, microbial production platforms possess great potential to produce a wide range of HMOs on a large scale.

## Figures and Tables

**Figure 1 molecules-28-01491-f001:**
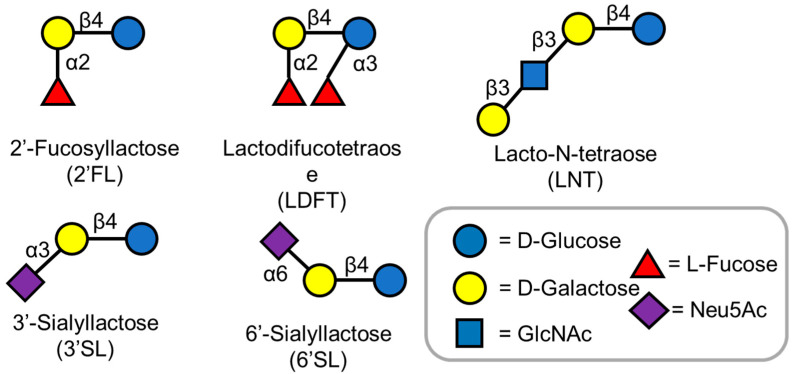
Structures of Human milk oligosaccharides described in this review. HMOs are made of five basic monosaccharides: D-glucose, D-galactose, L-fucose, N-acetyl-D-glucosamine (GlcNAc), and N-acetylneuraminic acid (Neu5A).

**Figure 2 molecules-28-01491-f002:**
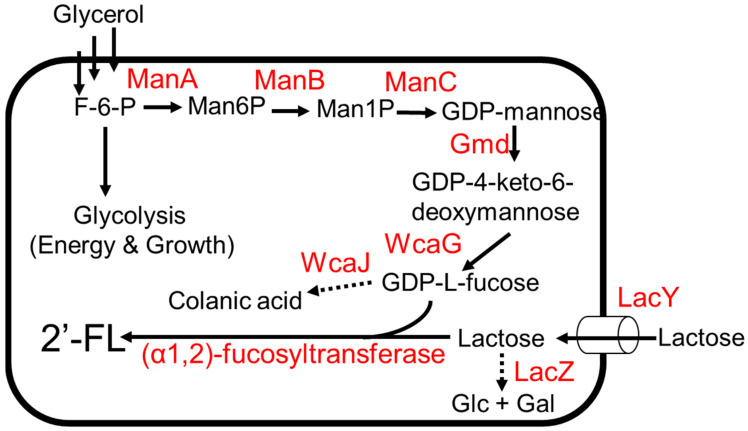
The 2′FL production from glycerol and lactose. Dotted arrows indicate competing pathways. Abbreviations: ManA, mannose-6-phosphate isomerase; ManB, phosphomannomutase; ManC, α-D-mannose-1-phosphate guanyltransferase; Gmd, GDP-D-mannose-4,6-dehyratase; WcaG, GDP-L-fucose synthetase; LacZ, β-galactosidase; WcaJ, UDP-glucose:undecaprenyl-phosphate glucose-1-phosphate transferase; LacY, lactose permease.

**Figure 3 molecules-28-01491-f003:**
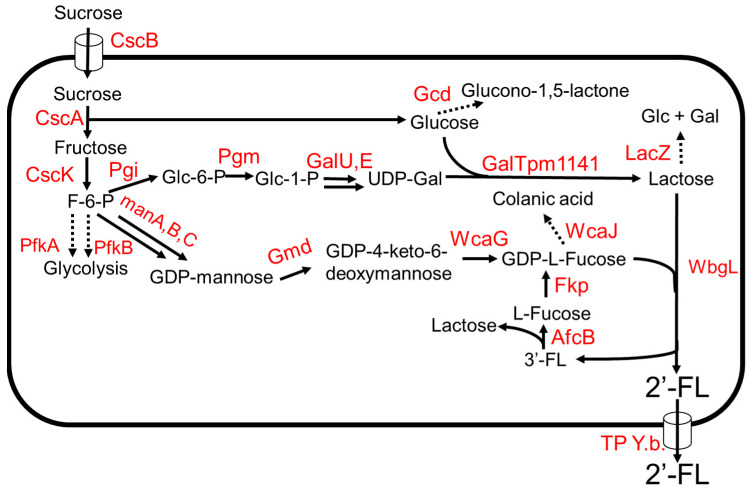
The 2′FL production from sucrose. Dotted arrows indicate competing pathways. Abbreviations: ManA, mannose-6-phosphate isomerase; ManB, phosphomannomutase; ManC, α-D-mannose-1-phosphate guanyltransferase; Gmd, GDP-D-mannose-4,6-dehyratase; WcaG, GDP-L-fucose synthetase; Pgi, glucose-6-phosphate isomerase; Pgm, phosphoglucomutase; GalU, UTP—glucose-1-phosphate uridylyltransferase; GalE, UDP-glucose 4-epimerase; WcaJ, UDP-glucose:undecaprenyl-phosphate glucose-1-phosphate transferase; Gcd, quinoprotein glucose dehydrogenase; LacZ, β-galactosidase; Fkp, bifunctional fucokinase/fucose-1-phosphate guanylyltransferase; AfcB, α1–3-fucosidase; GalTpm1141, b1–4-galactosyltransferase; WbgL, (α1,2)-fucosyltransferase; TP Y.b., sugar efflux transporter; CscB, sucrose permease; CscA, sucrose hydrolase; CscK, fructokinase; PfkA, 6-phosphofructokinase 1; PfkB, 6-phosphofructokinase 2.

**Figure 4 molecules-28-01491-f004:**
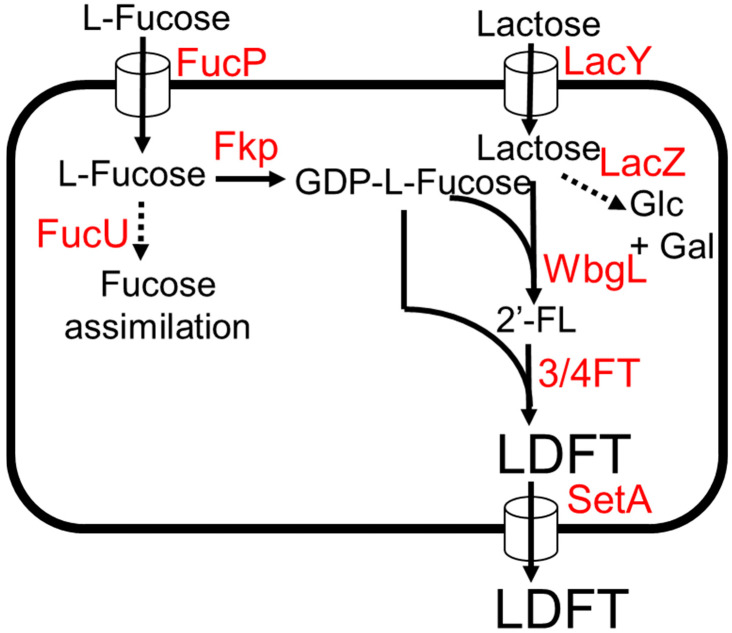
The LDFT production from L-fucose and lactose. Dotted arrows indicate competing pathways. Abbreviations: FucP, L-fucose:H^+^ symporter; Fkp, bifunctional fucokinase/fucose-1-phosphate guanylyltransferase; FucU, L-fucose mutarotase; LacY, lactose permease; LacZ, β-galactosidase; WbgL, (α1,2)-fucosyltransferase; ¾FT, (α1,3/4)-fucosyltransferase; SetA, sugar efflux transporter A.

**Figure 5 molecules-28-01491-f005:**
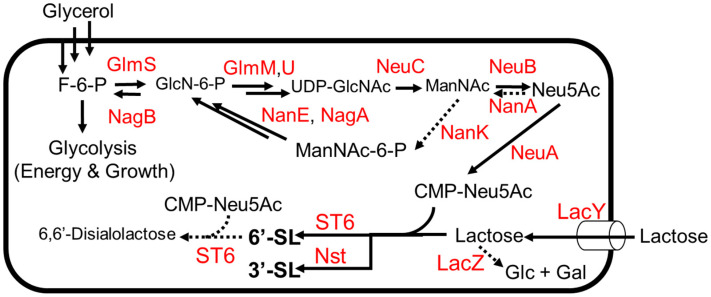
The 6′-SL and 3′-SL production from glycerol and lactose. Dotted arrows indicate competing pathways. Abbreviations: GlmS, L-glutamine—D-fructose-6-phosphate aminotransferase; GlmM, phosphoglucosamine mutase; GlmU, fused N-acetylglucosamine-1-phosphate uridyltransferase and glucosamine-1-phosphate acetyltransferase; LacZ, β-galactosidase; LacY, lactose permease; NagB, glucosamine-6-phosphate deaminase; NanE, *N*-acetylmannosamine-6-phosphate 2-epimerase; NagA, *N*-acetylglucosamine-6-phosphate deacetylase; NanK, *N*-acetylmannosamine kinase; NeuC, GlcNAc-6-phosphate 2 epimerase; NeuB, sialic acid synthase; NeuA, CMP-Neu5Ac synthetase; NanA, *N*-acetylneuraminate lyase; ST6, (α2,6)-Sialyltransferase; Nst, (α2,3) NeuAc transferase.

**Figure 6 molecules-28-01491-f006:**
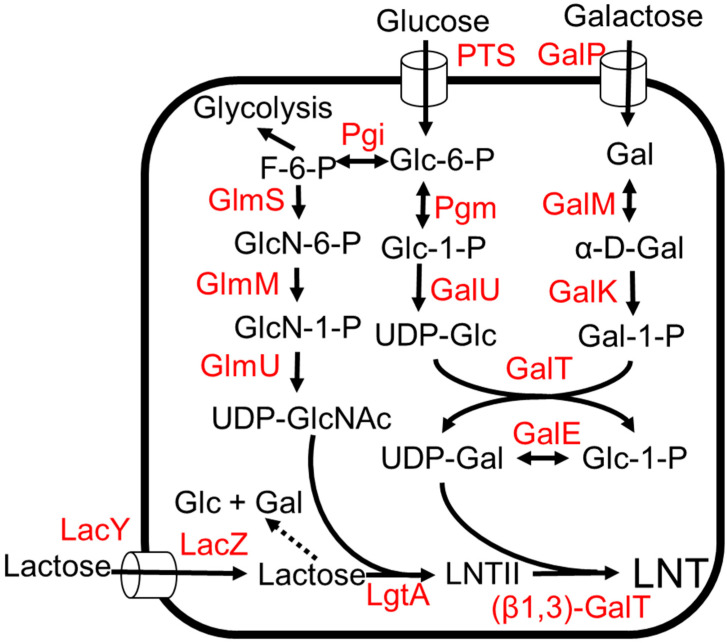
The LNT production from lactose and glucose or galactose. Dotted arrows indicate competing pathways. Abbreviations: PTS, phosphotransferase system; GlmS, L-glutamine—D-fructose-6-phosphate aminotransferase; GlmM, phosphoglucosamine mutase; GlmU, fused N-acetylglucosamine-1-phosphate uridyltransferase and glucosamine-1-phosphate acetyltransferase; Pgi, glucose-6-phosphate isomerase; Pgm, phosphoglucomutase; GalU, UTP—glucose-1-phosphate uridylyltransferase; GalT, galactose-1-phosphate uridylyltransferase; GalE, UDP-glucose 4-epimerase; GalK, galactokinase; GalM, galactose-1-epimerase; GalP, galactose:H^+^ symporter; LacY, lactose permease; LacZ, β-galactosidase; LgtA, (β1,3)-N-acetylglucosaminyltransferase; β1,3-GalT, (β1,3)-galactosyltransferase.

## Data Availability

No new data were created or analyzed in this study.

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
