# Peer review of "Microbial Production of Human Milk Oligosaccharides"

_molecules, 2023, doi:10.3390/molecules28031491_

Round 1

Reviewer 1 Report

Review to the manuscript of the review article entitled “Microbial Production of Human Milk Oligosaccharides” by Dileep Sai Kumar Palur et al. submitted to Molecules.

General remarks:

The manuscript of a review paper summarizes the advances on microbial production strategies of human milk oligosaccharides (HMOs) focusing on 5 most common and abundant short-chain HMO species. Recently, the metabolic engineering methods have been applied to construct whole-cell catalysts based on, e.g. E. coli, that could serve as feasible platforms to commercially produce food-grade HMOs for various applications. The topic is of very high importance as HMOs have enormous potential to be applied in food industry as functional food ingredients and infant food formulations.

The aims and scope of the manuscript is in accordance with that of the journal and the special issue where it was submitted.

The manuscript is adequately composed and structured. The focus is clearly set and most of the recent advances and future directions are highlighted in the review manuscript. Still the focus should be more clearly presented in the title and abstract (please see below, Specific comment 2).  The manuscript is written in an English language on academic style. The authors should carefully review the manuscript regarding the formatting (e.g. fonts; italics in species names, Latin phrases and N-linked sugars; uniform units).

Specific comments:

1)      Graphical abstract was not provided for the review. Comprehensive graphic abstract will give necessary background of and general overview of the topic: information on microbes and/or enzymes responsible for the reactions, substrates used and saccharides structures that are considered HMO-s.

2)      The manuscript focuses on five shorter HMO variants. The overall repertoire of naturally found HMOs is much more complex with hundreds of molecules with different structures. The whole-cell catalysts were producing some selected HMOs (and precursors for other HMOs) were summarized in review. The focus should be reflected more clearly in the title and in the abstract.

3)      Line 8 and line 40. “Antimicrobial” is not accurate to use as beneficial bacteria that are also microbes are induced by HMOs. The more correct would be “antagonistic to pathogens”.

4)      Line 29, 102, 117 and elsewhere. The presentation of linkage types is rather unconventional. The use of arrow and brackets (or alternatively brackets with comma) is encouraged.

5)      Fig. 1 and elsewhere in the text. 3’‐SL and 6’‐SL are commonly referred to as 3’‐sialyllactose and 6’‐sialyllactose, respectively. The name is used in many official documents including at EFSA. Some authors might use “sialolactose” but it should be presented as alternative name.

6)      Lines 64-74. The chemoenzymatic synthesis is combining chemical synthesis and enzymatic synthesis. OPME contains generally several enzymes in the system but omits chemical synthesis steps, therefore it is not part of chemoenzymatic synthesis. The section should be rephrased.

7)      Lines 112-113. B. fragilis mentioned here is a bacterium from a gut, not an eukaryotic organism. The sentence should be rephrased.

8)      Fig. 2-6. To facilitate the reading of the figures, the proteins should be marked differently (e.g. bold or different colour) to more easily distinguish them from reaction intermediates and products. If possible, the species name where the pathway was identified/constructed should be presented in the figure legend.

9)      Line 284. There is a mistake in the repeating unit (should be β instead of b).

10)   Many of the presented strategies involve the use of Gram-negative bacteria especially E. coli. The potential drawback of outer-membrane lipopolysaccharides as contaminants in the production should be discussed.

Author Response

We thank the reviewers for their comments. We have revised the manuscript to address the issues outlined. We have provided the revised manuscript in which all modifications have been highlighted in yellow.

The manuscript of a review paper summarizes the advances on microbial production strategies of human milk oligosaccharides (HMOs) focusing on 5 most common and abundant short-chain HMO species. Recently, the metabolic engineering methods have been applied to construct whole-cell catalysts based on, e.g. E. coli, that could serve as feasible platforms to commercially produce food-grade HMOs for various applications. The topic is of very high importance as HMOs have enormous potential to be applied in food industry as functional food ingredients and infant food formulations.

The aims and scope of the manuscript is in accordance with that of the journal and the special issue where it was submitted.

The manuscript is adequately composed and structured. The focus is clearly set and most of the recent advances and future directions are highlighted in the review manuscript. Still the focus should be more clearly presented in the title and abstract (please see below, Specific comment 2).  The manuscript is written in an English language on academic style. The authors should carefully review the manuscript regarding the formatting (e.g. fonts; italics in species names, Latin phrases and N-linked sugars; uniform units).

--- Thank you for your encouraging remarks.

Specific comments:

  • Graphical abstract was not provided for the review. Comprehensive graphic abstract will give necessary background of and general overview of the topic: information on microbes and/or enzymes responsible for the reactions, substrates used and saccharides structures that are considered HMO-s.

--- Thank you for the suggestion. However, our understanding is that this journal does not use graphical abstract. Thus, we did not attach graphic abstract to the revised manuscript. If the editor believes graphical abstract is required, we would be happy to provide it.

  • The manuscript focuses on five shorter HMO variants. The overall repertoire of naturally found HMOs is much more complex with hundreds of molecules with different structures. The whole-cell catalysts were producing some selected HMOs (and precursors for other HMOs) were summarized in review. The focus should be reflected more clearly in the title and in the abstract.

--- Thank you for pointing it out. We focused on these five particular HMOs that have been approved by the U.S. FDA for their commercialization and several companies are currently developing their industrial scale production methods. We clarify it in the abstract and text (line 16-17 & line 98-100).

  • Line 8 and line 40. “Antimicrobial” is not accurate to use as beneficial bacteria that are also microbes are induced by HMOs. The more correct would be “antagonistic to pathogens”.

---- Thank you for the suggestion. We’ve corrected it (line 44-45 and other places).

  • Line 29, 102, 117 and elsewhere. The presentation of linkage types is rather unconventional. The use of arrow and brackets (or alternatively brackets with comma) is encouraged.

---- Thank you for the suggestion. We’ve corrected it by using brackets with commas as suggested.

  • 1 and elsewhere in the text.3’‐SL and 6’‐SL are commonly referred to as 3’‐sialyllactose and 6’‐sialyllactose, respectively. The name is used in many official documents including at EFSA. Some authors might use “sialolactose” but it should be presented as alternative name.

---- Thank you. We have changed the name to “sialyl” (line 228 and other places).

  • Lines 64-74. The chemoenzymatic synthesis is combining chemical synthesis and enzymatic synthesis. OPME contains generally several enzymes in the system but omits chemical synthesis steps, therefore it is not part of chemoenzymatic synthesis. The section should be rephrased.

---- Thank you for the suggestion. We have separated them into two different paragraphs.

  • Lines 112-113.  fragilismentioned here is a bacterium from a gut, not an eukaryotic organism. The sentence should be rephrased.

---- Thank you for pointing it out. We have rephrased the sentence. It now reads: “The salvage pathway, generally found in eukaryotes, has also been recently discovered in the gut bacterium Bacteroides fragilis [39].” (Line127-128)

  • 2-6. To facilitate the reading of the figures, the proteins should be marked differently (e.g. bold or different colour) to more easily distinguish them from reaction intermediates and products. If possible, the species name where the pathway was identified/constructed should be presented in the figure legend.

--- Thank you for the suggestion. We used red color for the enzymes in the figures. Species names are not included because adding them would make the figures too busy. Additionally, many of the enzymes described in this review are found in multiple species.

  • Line 284. There is a mistake in the repeating unit (should be β instead of b).

--- Thank you for pointing it out. We have fixed it.

  • Many of the presented strategies involve the use of Gram-negative bacteria especially  coli. The potential drawback of outer-membrane lipopolysaccharides as contaminants in the production should be discussed.

--- Thank you for the suggestion. We have added discussion about it in the Challenges section. It now reads: “When considering the application of whole-cell biocatalysts in the production of food additives, food safety must be thoroughly considered. Food products based on Gram-negative bacteria such as Escherichia coli increase the likelihood of lipopolysaccharide contamination [67]. Lipopolysaccharides are a structural component of bacterial cell and are also known as endotoxins [68]. Bacterial lipopolysaccharides have been associated with a number of diseases, including liver damage, neurological degeneration, gut inflammation, and diabetes [67]. Purification of HMOs produced from E. coli to remove these endotoxins is a crucial safeguard and will remain a major hurdle for the commercialization of these microbial production platforms [68].” (Line 333-341)

Reviewer 2 Report

1. Keywords: The first letter should be capitalized.

2. Introduction: The logical progression between the first paragraph and the second paragraph needs further adjustment. “Carbohydrates are the most abundant biomolecules on earth and also are the primary energy source for diverse organisms” should be deleted.

3. In Line 41, “The mechanistic understanding of these bioactive properties is not well understood …... there is emerging need to synthesize these complex molecules”, why isn't further exploration of the mechanism a priority? The expression of this part needs to be revised. The need for synthesis can be elicited by highlighting the importance of HMOs and their current low yield. Anything unrelated to this paper need not be mentioned.

4. The section 2 and section 3 have the same title, it needs to be amended.

5. In Line 87, “In this review, we summarize the latest progress in ……”, this sentence is usually at the end of the Introduction. The section 2 can be merged with the Introduction.

6. In Line 184, the format of this paragraph is inconsistent with others.

7. In Line 261, the section title is sorted incorrectly and should be changed to 3.4.

8. Why does this paper focus on the structure and production of these five HMOs? It can be added to explain the several major compositions of HMOs in Introduction.

9. In Line 319, there is a period missing at the end of the sentence. It needs to be added.

Author Response

We thank the reviewer for their comments. We have revised the manuscript to address the issues outlined. We have provided the revised manuscript in which all modifications have been highlighted in yellow.

  1. Keywords: The first letter should be capitalized.

 --- We have fixed them.

  1. Introduction: The logical progression between the first paragraph and the second paragraph needs further adjustment. “Carbohydrates are the most abundant biomolecules on earth and also are the primary energy source for diverse organisms” should be deleted.

 --- Thank you for your suggestion. We added a connection sentence to both paragraphs (Line 33-34 & 41-42).

  1. In Line 41, “The mechanistic understanding of these bioactive properties is not well understood …... there is emerging need to synthesize these complex molecules”, why isn't further exploration of the mechanism a priority? The expression of this part needs to be revised. The need for synthesis can be elicited by highlighting the importance of HMOs and their current low yield. Anything unrelated to this paper need not be mentioned.

 --- Thank you for the suggestion. We have rephrased the sentences to explain why study of the bioactivities of these HMOs are important. It now reads: “HMOs possess prebiotic, immunomodulatory, antagonistic properties towards pathogens and can also serve as a key support for neurocognitive development of infants [8,9,10]. These and other potential benefits of HMOs make them attractive research targets for preventing or treating diseases in both children and adults [8]. The bioactive properties of HMOs have prompted efforts to elucidate the mechanisms of the individual HMO's actions [8,9,10], but the sources of HMOs for such studies are limited. Due to limited access to these HMOs, there is an emerging need to synthesize these complex molecules [10].”

  1. The section 2 and section 3 have the same title, it needs to be amended.

 --- Thank you for pointing it out. The title of the section 3 has been changed to “HMO Production by Microbial Cell Factories”.

  1. In Line 87, “In this review, we summarize the latest progress in ……”, this sentence is usually at the end of the Introduction. The section 2 can be merged with the Introduction.

 --- Thank you for the suggestion. However, we believe that there is no problem with this sentence in this location, so we will leave it as it is.

  1. In Line 184, the format of this paragraph is inconsistent with others.

 --- We could not see any inconsistency in the paragraph. Please specify it if it is a critical problem.

  1. In Line 261, the section title is sorted incorrectly and should be changed to 3.4.

 --- Thanks. It was fixed.

  1. Why does this paper focus on the structure and production of these five HMOs? It can be added to explain the several major compositions of HMOs in Introduction.

 --- We focused on the microbial production of these five particular HMOs that are approved by the U.S. FDA for their commercialization and several companies are currently developing their industrial scale production methods. We clarify it in the abstract and text (line 16-17 & line 98-100).

  1. In Line 319, there is a period missing at the end of the sentence. It needs to be added.

--- On our version, the sentence and the surrounding sentences have periods. Any oversights will be corrected during proofreading.

Round 2

Reviewer 2 Report

The authors have addressed all my comments in this revision.